# Tau beyond Tangles: DNA Damage Response and Cytoskeletal Protein Crosstalk on Neurodegeneration

**DOI:** 10.3390/ijms25147906

**Published:** 2024-07-19

**Authors:** Megumi Asada-Utsugi, Makoto Urushitani

**Affiliations:** Department of Neurology, Molecular Neuroscience Research Center, Shiga University of Medical Science, Otsu 520-2192, Shiga, Japan; utsumegu@belle.shiga-med.ac.jp

**Keywords:** DDR, tau, DSB, cytoskeletal protein, neurodegenerative diseases

## Abstract

Neurons in the brain are continuously exposed to various sources of DNA damage. Although the mechanisms of DNA damage repair in mitotic cells have been extensively characterized, the repair pathways in post-mitotic neurons are still largely elusive. Moreover, inaccurate repair can result in deleterious mutations, including deletions, insertions, and chromosomal translocations, ultimately compromising genomic stability. Since neurons are terminally differentiated cells, they cannot employ homologous recombination (HR) for double-strand break (DSB) repair, suggesting the existence of neuron-specific repair mechanisms. Our research has centered on the microtubule-associated protein tau (MAPT), a crucial pathological protein implicated in neurodegenerative diseases, and its interplay with neurons’ DNA damage response (DDR). This review aims to provide an updated synthesis of the current understanding of the complex interplay between DDR and cytoskeletal proteins in neurons, with a particular focus on the role of tau in neurodegenerative disorders.

## 1. Introduction

The human genome is constantly subjected to various forms of DNA damage, but the brain possesses intricate mechanisms to repair this damage and maintain genomic stability. DNA damage occurs not only in somatic cells but also in neurons, and precise repair of this damage is crucial for maintaining brain homeostasis [1,2,3]. Neuronal activity can induce transient DNA double-strand breaks (DSBs) in neurons, which are usually repaired within minutes without compromising neuronal function [1,4]. However, studies have shown that DNA damage accumulates in the nuclei of aging cells and neurons in neurodegenerative diseases [5,6,7]. Since neurons are post-mitotic cells arrested in the G0 phase, except for those in certain brain regions, it is hypothesized that they may not be able to repair DNA damage using homologous recombination (HR) as efficiently as somatic cells. Although DSB repair is considered a critical function in neurons, the precise mechanisms of the DNA damage response (DDR) in these cells remain elusive.

One of the fundamental functions of the central nervous system is the formation and storage of memories. Synaptic plasticity, a key process in the transition from recent to remote memory, dynamically modulates information transfer efficiency and synaptic connections by altering spine morphology by activating glutamate receptors in the hippocampus. These changes in spine morphology are tightly regulated by the remodeling of the cytoskeletal proteins actin and microtubules [8,9,10,11]. A recent study has revealed novel findings suggesting that memory formation and persistence require hippocampal neuron-specific inflammatory signaling, which mediates DSB generation, DDR, and the formation of primary cilia composed of microtubules [12]. Episodic memory, which is impaired in Alzheimer’s disease (AD) [13], is particularly affected by these processes. The interplay between DDR and cytoskeletal proteins in neurons is a remarkable phenomenon that warrants further investigation. Neurofibrillary tangles (NFTs), one of the hallmark pathologies of AD, are caused by the hyperphosphorylation of the microtubule-associated protein tau (MAPT; tau) and its accumulation in neurons [14]. The authors have previously reported that tau is involved in microtubule-mediated DDR in neurons [15]. In this review, we present the relationship between DDR in neurons and neurodegenerative diseases based on our findings and recent advances in the field. By exploring the intricate connections between these processes, we aim to shed light on potential therapeutic targets and strategies for neurodegenerative disorders.

## 2. Development and Repair of DNA Damage in Neurons

Loss of genomic integrity is strongly associated with aging and neurodegeneration [16,17]. Many age-related neurodegenerative diseases exhibit both an accumulation of DNA damage and decreased efficiency of DNA damage repair [5,6,7]. Among these lesions, DSBs are the most toxic and have been reported to cause various phenotypes of aging, including senescence, mutation, and cell death. In differentiated neurons, protecting the integrity of genetic regions is critical for maintaining a healthy brain environment, and unrepaired DSBs in neurons can ultimately lead to cell death. Furthermore, incorrect repair can induce severe mutations, such as deletions, insertions, and chromosomal translocations, risking significant loss of genetic information [18,19].

Interestingly, there are many reports of DSB formation in neuronal nuclei in an activity-dependent manner. Activity-dependent intracellular calcium influx via N-methyl-D-aspartate (NMDA) receptors activates the phosphatase calcineurin, which dephosphorylates residues S1509 and S1511 of Topoisomerase IIβ (Top2B), thereby activating its DNA cleavage activity and inducing DSB formation. Top2B activation also generates DSBs in the promoters of early response genes in neurons and promotes their transcription [20]. Additionally, when wild-type mice are subjected to the fear conditioning test, a paradigm used to validate the formation of recent memory, Top2B is dephosphorylated at S1509 and S1511 in hippocampal CA1 neurons, suggesting that calcineurin also regulates Top2B-mediated DSB formation after physiological neural activity [20]. Furthermore, DSBs increase in the brains of wild-type mice subjected to novel exploratory activities but decrease when they are returned to their home cage. Exposure of the wild-type mouse brain to amyloid-β also increases DSBs [3]. Single-neuron studies in zebrafish have reported that DSBs accumulate during waking hours, while during sleep, increased chromosome movement is observed and DSBs decrease [21]. Collectively, these findings suggest that DSB generation in neurons is a neural activity-dependent physiological phenomenon, and the brain can quickly repair routinely generated DSBs.

## 3. DSB Repair System

Two main DSB repair pathways are known from previous DNA repair studies. One is nonhomologous end joining (NHEJ), a simple and rapid repair pathway that links the ends of a resulting DNA break [22]. The other is homologous recombination (HR), which repairs the damage while restoring the original sequence by initiating a recombination reaction between sister chromatids that are free of DNA damage and created after DNA replication. However, since neurons do not have sister chromatids, they likely use NHEJ for DSB repair. Recent studies have shown that in G1 cells without sister chromatids, apart from canonical NHEJ (cNHEJ), which is the major NHEJ pathway, alternative NHEJ (alt-NHEJ) and transcription-associated end joining (TA-EJ) are also involved in DSB repair [23,24] (Figure 1). Future studies are expected to clarify which NHEJ pathway neurons preferentially use.

## 4. DDR and Cytoskeletal Proteins

Cytoskeletal proteins play a crucial role in DSB repair. Reports in yeast, Drosophila, and mouse cells indicate that when severe DSBs occur in heterochromatin, nuclear actin polymerizes, and myosin motor proteins transport the DSB site along the polymerized actin filaments to the nuclear periphery for repair [25]. In the case of tubulin, it has been reported that upon DSB formation, the nuclear membrane is invaded, and repair factors are recruited to the DSB site [2].

### 4.1. DDR and Nuclear Membrane

Nuclear invaginations are often observed in aged neurons and neurons from patients with frontotemporal dementia. Some reports suggest that these invaginations contain microtubules, nuclear membrane complexes, and phosphorylated tau that have invaded the nuclear membrane [26]. In the most recent study, Shokrollahi et al. reported that the linker of the nuclear skeleton and cytoskeleton complex (LINC) and nuclear pore complex (NPC) proteins cooperate with the nuclear lamina and kinesins KIF5B and KIF13B to form nuclear envelope tubules to DSB sites in human cells. When DSB is induced, the DDR kinases ATM, DNAPKcs, and ATR cross with ATAT1-dependent lysine 40 acetylation of α-tubulin. The modified microtubules cooperate with the plus-end directed motor proteins kinesin-1 KIF5B, kinesin-3 KIF13B, LINC, and NPC complexes. In neurons, the kinesin superfamily is a motor protein that transports cargo proteins over microtubules and is responsible for the transport of various proteins, organelles, and mRNAs in the cell [27,28]. Microtubules are pressed against the cytoplasmic face of the nuclear membrane and delivered to the DSB site via the resulting nuclear invasion tubular infiltrate. This tubulin-dependent nuclear invasion tubule partially promotes DSB binding, partially stabilizes microtubules, and helps direct DSBs toward the cleavage point. Because many DSB repair factors are present around the nuclear membrane, tubulin-dependent nuclear membrane entry tubes transport the nuclear membrane to the DSB and facilitate repair. Nuclear membrane entry tubes confine DSB ends within repair centers and provide strong support that promotes DNA repair protein binding and reconnection of cleaved ends [27]. Reports of phosphorylated tau entering the nucleus along with microtubules in the nuclei of FTD patient brains [26] and our results of tau accumulation around the nuclear membrane during DSB induction [15] suggest that tau may be involved in tubulin-dependent nuclear membrane entry tract dynamics in neurons, but the details are unclear.

Moreover, they observed increased 53BP1 foci following knockdown of LINC subunits SUN1 and SUN2 [29]. It has also been reported that phosphorylated tau accumulates in NPCs in the AD brain, inhibiting nucleocytoplasmic transport [30]. Jovasevic V. et al. also reported increased nuclear membrane disruption and extranuclear release of γH2Ax with increased DSB after a contextual fear conditioning test in wild-type mice [12]. Mutations in the *LMNA* gene, which encodes the nuclear membrane proteins lamins A and C, cause several human diseases (laminopathies), including dilated cardiomyopathy, Emery–Dreifuss muscular dystrophy, *LMNA*-related congenital muscular dystrophy, and Hutchinson–Gilford premature aging syndrome [31]. Earle A.J. et al. demonstrated that mechanical forces in *LMNA* mutant skeletal muscle cells cause myonuclear damage, nuclear membrane rupture, and DNA damage [32]. Rupture of the nuclear membrane causes DNA damage, including mislocalization of DNA repair factors and exposure of DNA to cytoplasmic exonucleases. Differentiated skeletal muscle cells downregulate DNA repair factors and are less efficient at DNA repair than undifferentiated myoblasts.

Furthermore, recent findings suggest that DNA damage and activation of the DDR pathway may be the pathogenic mechanisms of myopathy [33]. Kirby T.J. et al. showed increased p53 stabilization and activity in mechanically injured *LMNA* mutant myocytes. Furthermore, they showed that a chronic increase in p53 signaling is required to induce dysfunction in wild-type myocytes [34].

Thus, nuclear membrane depression may be due to the disruption of cytoskeletal proteins that provide scaffolding for the nuclear membrane, resulting from aging or DSB repair failure.

### 4.2. DDR and Memory Formation

Memory formation involves two main mechanisms. One is synaptic plasticity, in which activating glutamate receptors in the hippocampus alters spine morphology, modulating information transmission efficiency and synaptic connections. Cytoskeletal protein dynamics regulate these changes in spine morphology. The other mechanism is the expression of a group of early response genes induced by neural activity. Furthermore, it has been reported that DSBs generated after neural activity induce the expression of early response genes such as *c-fos*, *Arc*, *Egr-1*, *Npas4*, etc. [12,35,36,37]. RNA sequencing of hippocampal CA1 neurons during the transition from recent to remote memory formation has shown increased expression of actin- and tubulin-related genes and a group of genes involved in the formation of primary cilia, which are composed of microtubules [8]. These findings suggest that DSB formation and cytoskeletal protein dynamics in neurons are closely related to memory formation. *Npas4* (neuronal PAS domain-containing protein 4), one of the earliest genes and transcription factors, has been shown to promote synaptogenesis in an activity-dependent manner in excitatory neurons in the cortex and hippocampus [38]. More recently, a new function of *Npas4* has been reported, in which it binds to the NuA4 histone acetyltransferase complex at DSB sites and recruits DNA repair factors such as MRE11, RAD50, and NBS1 [36]. Thus, cytoskeletal protein dynamics and DDR are closely intertwined in neurons.

## 5. Disruption of DNA Damage Repair and Neurodegenerative Diseases

DSB accumulation has been identified in several neurodegenerative diseases. In mouse models of neurodegeneration, including the Tau P301S and P301L mutant tauopathy models, the CK-p25 model, and the hAPP-J20 amyloid pathology model, increased DSBs have been observed early in the pathogenesis of the disease [17,39]. Additionally, the toxicity of DSBs seen in early Alzheimer’s disease (AD) may act as an early trigger [40]. The accumulation of DSBs in neurons in the early stages of AD also leads to genomic structural changes. Single-cell whole-genome sequencing of neurons in DNA repair-deficient diseases with progressive neurodegeneration (Cockayne syndrome, xeroderma pigmentosum, and ataxia–telangiectasia) has also revealed an age-dependent increase in somatic cell deletions resulting from DNA repair deficiency [41].

### 5.1. Neurodegenerative Diseases and Genomic Structural Changes

Recent 3D genome studies have revealed that chromatin interactions, including chromatin loops, form self-interacting topologically associating domains (TADs) and spatially segregate into transcriptionally active and repressive compartments in the nucleus [42]. This genomic architecture is closely related to essential biological functions such as gene expression, DNA replication, genome stability, DNA repair, development, and the cell cycle. It has also been reported that neurons with accumulated DSBs exhibit changes in the 3D structure of the genome, resulting in gene fusion and genomic reorganization [43]. These results suggest that the disruption of genomic stability by DSBs and changes in the 3D structure of the genome are associated with pathological stages in the progression of neurodegenerative diseases. In neurodevelopmental disorders such as Fragile X syndrome and Down syndrome, disruption of the 3D structure of the genome has also been demonstrated [44,45]. In late-stage AD (Braak stages 5–6), characterized by the presence of neurofibrillary tangles (NFTs) containing hyperphosphorylated tau, gene fusions are significantly enriched in excitatory neurons [43]. Thus, DSB-mediated senescence-like states in neurons can also disrupt the nuclear 3D genome structure in the nucleus in neurodegenerative diseases.

### 5.2. Presence of DSBs in the AD Brain

AD develops following the accumulation of amyloid-β and tau, ultimately leading to neuronal cell death. NFT formation is deeply involved in neuronal death, and an integrated understanding of NFT formation is essential for preventing the onset of AD. Many studies have shown an increase in DSB in AD brains [15,40,46,47]. Our previous studies found a significant increase in the DSB marker γH2AX, a phosphorylated histone H2AX variant at serine 139, in the hippocampus and temporal cortex of AD autopsy brains compared to the non-neurodegenerative group. Analysis of the cell types harboring DSBs revealed their presence in GFAP-positive astrocytes in the hippocampus, neurons in the temporal cortex, and oligodendrocytes in the white matter. In microglia, only slight co-localization with DSBs was observed in any region. Thus, the presence of DSBs was confirmed in neurons, astrocytes, and microglia in various areas of the AD brain [15].

### 5.3. Relationship between DDR and Tau

Tau may be associated with DDR because of its DNA-binding capacity and reports on chromatin remodeling [48,49]. In our study results, when DSBs were induced in primary mouse cortical neurons using etoposide, a topoisomerase II inhibitor, non-phosphorylated tau (recognized by the Tau1 antibody, which detects tau dephosphorylated at S195, S198, S199, and S202) accumulated around the nuclear membrane and in the cytoplasm within 30 min of DSB induction, and its binding to microtubules also increased. Twenty-four hours after DSB induction, phosphorylated tau accumulated at the perinuclear membrane and in the cytoplasm, enhancing neuronal cell death [15]. The cytoplasmic enzyme GSK3β, a tau kinase that is increased in AD, is reported to translocate to the nucleus upon DSB induction, suggesting that DSBs activate a group of tau phosphorylation regulators [50]. Previous studies have shown that neuronal activity increases the localization of Top2B and calcineurin to the nuclear periphery, and induced DSB sites also localize to the nuclear membrane [20], suggesting a link to our observed DSB-induced increase in tau localization to the perinuclear membrane. To investigate the role of tau accumulation in the DDR, we performed a knockdown of endogenous mouse tau in primary mouse cortical neurons. Reducing endogenous mouse tau increased γH2AX levels following short-term DSB induction by etoposide. It has been reported that hippocampal γH2Ax is also increased in tau knockout mice [49]. However, under prolonged DSB accumulation in neurons, the knockdown of endogenous tau reduced γH2AX levels. These results suggest that tau may have a protective role in early DSB repair, but under conditions of persistent DSB accumulation, it may promote tau phosphorylation and neuronal cell death. Since neurons are thought to be unable to use homologous recombination, it is suggested that they may use NHEJ repair, which is a speedy repair. These results suggest that tau is involved in NHEJ repair.

### 5.4. Microtubule Polymerization and DDR

Tau is a microtubule-binding protein that localizes mainly to the axons of neurons and regulates microtubule polymerization through phosphorylation. Tau is physiologically phosphorylated and is most highly phosphorylated during the fetal stage in the mouse brain. In pathological conditions such as AD, tau is hyperphosphorylated and forms NFTs in neurons, which aggregate abnormally and cause neuronal cell death [51,52,53]. A study examining the effects of tau and DSBs on tubulin polymerization in primary mouse neurons reported the appearance of NFT-like neurons with phosphorylated tau accumulation and increased cleaved caspase-3-positive neuronal cell death after DSB induction following microtubule polymerization inhibition [15]. These findings indicate that the combination of microtubule depolymerization and DSB accumulation promotes the accumulation of phosphorylated tau in neurons. Additionally, we quantified amyloid-β in the supernatants of primary mouse neuronal cell cultures treated with varying concentrations of etoposide and found no significant changes, suggesting that DSB accumulation does not affect amyloid precursor protein (APP) cleavage [15]. These results indicate that DSB repair impairment, microtubule depolymerization, and phosphorylated tau accumulation are heavily involved in the early stages of AD pathogenesis (Figure 2). However, the mechanism of NFT formation remains unclear, and no model has yet recapitulated NFTs from endogenous tau. Elucidating the regulation of tau and its physiological mechanisms is crucial for understanding AD and deciphering the pathophysiology of tauopathies, which are pathologically characterized by tau aggregation. The mechanism of neuronal DSB repair remains to be fully elucidated.

### 5.5. DDR and Pathogenic Proteins in Neurodegenerative Diseases

Tauopathies, such as AD, are characterized by the accumulation of phosphorylated tau in neurons and glial cells. Tau is a microtubule-binding protein that localizes mainly to axons or dendrites in neurons, where it stabilizes microtubules [52]; it is also localized in the nucleus, primarily localized in the nucleolus [57,58,59]. In breast cancer cells, tau has been reported to bind to chromatin and contribute to chromatin integrity [54,56], and it also chaperones the trafficking of the tumor suppressor p53-binding protein 1 (53BP1), a DSB repair factor, to DSB sites [55]. These results suggest that tau may also have a protective role in neuronal DDR. For amyloid-β, a major component of amyloid plaques and a typical AD pathology, it has been reported that administration of amyloid-β peptide to neurons and mice increases DSBs [3,47]. In addition, the accumulation of DSBs has been reported at an early stage in AD model mouse brains where amyloid-β accumulates [40].

Synucleinopathies, such as Parkinson’s disease, are characterized by the accumulation of phosphorylated α-synuclein in neurons and glial cells. α-Synuclein is mainly localized to the presynaptic terminals of neurons and is thought to be involved in synaptic regulation and plasticity. Synuclein is also found to localize to the nucleus and has NEHJ repair capacity [60]. Conversely, overexpression of α-synuclein causes cellular senescence and activates the p53 pathway and NHEJ. However, expression of MRE11, a critical factor in the DSB repair system, is reduced, suggesting incomplete induction of the repair pathway. Neuropathological examination of α-synuclein transgenic mice showed that phosphorylated α-synuclein and DNA damage accumulate early in the presymptomatic phase, and DSB levels and cellular senescence markers are elevated [61].

Polyglutamine diseases, such as Huntington’s disease and spinocerebellar ataxia, are caused by the accumulation of gene products containing expanded repeat sequences in neuronal nuclei [62]. Huntingtin, the gene responsible for Huntington’s disease, has been reported to bind the NHEJ factor Ku70 [63,64]. SCA3 is the most common form of spinocerebellar ataxia worldwide, caused by a polyglutamine (polyQ) repeat at the Ataxin-3 (ATX3) C-terminus. ATX3 has been shown to interact with polynucleotide kinase 3′-phosphatase (PNKP), the mediator of DNA damage checkpoint protein 1 (MDC1), checkpoint kinase 1 (Chk1), huntingtin (HTT), Ku70, DNA-PKcs, 53BP1, and p97 [65]. Gall-Duncan T. et al. reported increased expression of the ssDNA-binding complexes canonical replication protein A (RPA1, RPA2, RPA3) and Alternative-RPA in the brains of patients with Huntington’s disease and spinocerebellar ataxia type 1 (SCA1). Furthermore, overexpression of RPA in SCA1 model mouse brains reduced ATXN1 aggregation and DNA damage, restored neuronal morphology and motor phenotype, and suppressed CAG repeat expansion [66].

The proteins implicated in amyotrophic lateral sclerosis (ALS), including TAR DNA-binding protein 43 (TDP-43), fused in sarcoma (FUS), superoxide dismutase 1 (SOD1), and chromosome 9 open reading frame 72 (C9orf72) repeat peptides, are also related to the DDR. TDP-43 is an RNA-binding nuclear protein that stabilizes RNA [67,68]. Additionally, a physiological function of TDP-43 related to the DDR is the recruitment of the NHEJ factor XRCC4/DNA ligase IV complex to DSB sites [69]. Furthermore, TDP43 binds to the NHEJ factor Ku70 in HEK293 cells. Knockdown of TDP-43 in HeLa cells enhances R-loop structure (Figure 1) and DSBs [70]. FUS is also involved in the transport of mitochondrial DNA ligase IIIα and base excision repair (BER) [71]. SOD1 protects against DNA damage caused by oxidative stress [70]. In HEK293 cells, C9orf72 has been reported to interact with the NHEJ factor DNA-dependent protein kinase (DNA-PK) complex after DSB induction [72].

As such, many of the proteins implicated in neurodegenerative diseases have DDR-related functions in the nucleus (Table 1). It is suggested that DDR may be present upstream as a common neurodegenerative disease mechanism.

## 6. Conclusions

Recent groundbreaking studies on neurodegenerative diseases and DDR have been reported, shedding light on a previously unrecognized link. We proposed the novel idea of DDR in NFT-mediated neuronal death in AD. We also suggested that DDR is involved in early DSB repair via microtubule polymerization, indicating a novel physiological function of DDR in neurons. These findings have the potential to revolutionize the treatment of AD and other tau diseases that pathologize tau aggregation, leading to the development of new therapeutic strategies. These findings may provide a different approach to conventional AD treatment. Several important questions remain, i.e., “Is tau involved in tubulin-dependent nuclear membrane invasion tract dynamics in neurons?”; “By what mechanism is tau converted from DSB repair function to toxic phosphorylated aggregates?”; “Does inhibition of DSB repair lead to NFT formation?”; and “What is the relationship between the formation of phosphorylated tau aggregates in glial cells and DSBs in other tauopathies?”. We hope to explain the pathophysiology of tauopathy from a DDR perspective by elucidating the physiological functions of tau, which are not yet fully understood.

## Figures and Tables

**Figure 1 ijms-25-07906-f001:**
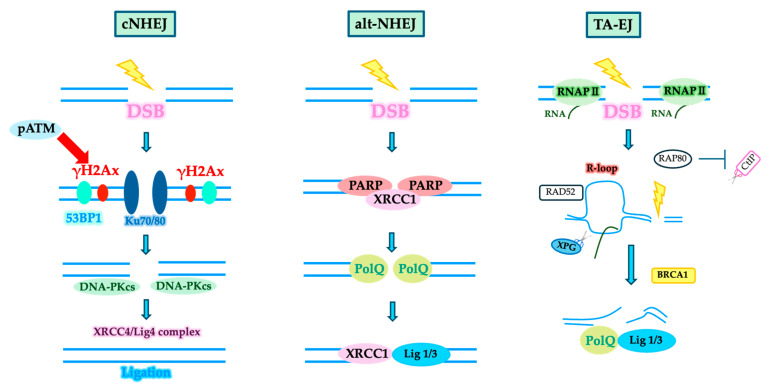
Various NHEJ pathways involved in DNA double-strand break repair (DSB). Canonical NHEJ (cNHEJ) is the primary NHEJ pathway in which phosphorylated ATM phosphorylates histone H2AX at serine 139 (γH2AX) in the vicinity of DSBs, and downstream repair enzymes bind to the break ends. Alternative NHEJ (alt-NHEJ) is an alternative pathway to cNHEJ that utilizes different repair enzymes. Transcription-associated end joining (TA-EJ) is a transcription-coupled end-joining pathway in which R-loop structures (hybrid DNA/RNA structures) near DSBs that arise in gene regions are protected by RAP80. Adapted from Shibata, 2017; Shibata and Jeggo, 2019; Yasuhara et al., 2022 [22,23,24]. For detailed molecular mechanisms, see the cited references.

**Figure 2 ijms-25-07906-f002:**
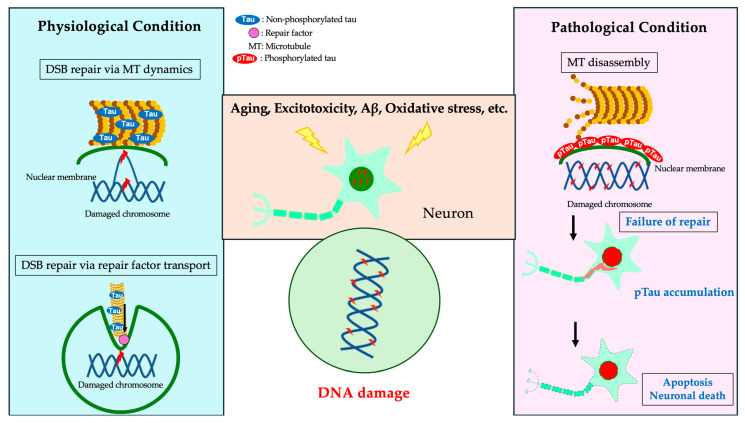
A schematic representation of a hypothesized DSB repair mediated by tau and microtubule dynamics. Under physiological conditions (**left panel**), non-phosphorylated tau (blue) binds to microtubules, promoting their polymerization and accumulation around the perinuclear membrane to support DNA damage repair. When DNA damage reaches excessive levels (**right panel**), tau phosphorylation is enhanced, and microtubule depolymerization occurs. The accumulation of phosphorylated tau (red) around the perinuclear membrane may suppress the movement of damaged chromosomes or inhibit the recruitment of DNA damage repair factors into the nucleus by inhibiting nuclear–cytoplasmic transport. Subsequently, insoluble phosphorylated tau accumulates in the cytoplasm and axons, leading to further microtubule depolymerization and neuronal death [15,54,55,56].

**Table 1 ijms-25-07906-t001:** DDR-related functions on pathogenic proteins of neurodegenerative diseases.

Pathogenic Proteins	Neurodegenerative Diseases	Subcellular Localization	DDR-Related Functions
MAPT (Tau)	AD, ALS, FTLD, PSP, CBD, PiD, AGD, CTE	Cytoplasm, Nucleus, Axon, Dendrite, Cell membrane	53BP1 transport (Breast cancer cell) [53] Chromatin remodeling (Breast cancer cell) [52,54] DSB repair on KD and KO [15,47]
Amyloid-β	AD	Cell surface	Enhancing of DSB [17,37,45]
TDP-43	ALS, FTLD, LATE, Perry disease, FOSMN	Nucleus, Cytoplasm, Mitochondria	XRCC4/Lig4 complex transport [65] Interaction with Ku70 (HEK293 cell) [66] KD enhances the R loops and DSB (HeLa cell) [66]
FUS	ALS, FTLD	Nucleus, Cytoplasm	mtDNA Ligase IIIα (mtLig3) transport BER [67]
C9orf72 repeat peptide	ALS, FTLD	Nucleus, Cytoplasm, Endsome, Lysosome, Axon	Interaction with DNA-PKcs complex [68]
SOD1	ALS, STAHP	Cytoplasm, Nucleus	DNA protection from oxdative damage [66]
HTT	Huntington disease	Nucleus, Cytoplasm, Early endosome	Interaction with Ku70 [61,62]
α-Synuclein	PD	Cytoplasm membrane, Nucleus, Synapse	Overexpression decreases NHEJ and MRE11 expression [58,59]
ATX3	SCA3	Nucleus, Nucleus matrix, Lysosome membrane	Interaction with PNKP, MDC1, Ku70, Chk1, HTT, DNA-PKcs, 53BP1 and p97 [63]
ATX1	SCA1	Nucleus, Cytoplasm	Increased expression of RPA1, RPA2, RPA3 and Alternative-RPA [64]

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
