# Peer review of "Tau Beyond Tangles: DNA Damage Response and Cytoskeletal Protein Crosstalk on Neurodegeneration"

_ijms, 2024, doi:10.3390/ijms25147906_

Round 1
Reviewer 1 Report
Comments and Suggestions for Authors
The review titled ‘Tau Beyond Tangles: Unveiling Its Role in Neuronal DNA Damage Response and Implications for Alzheimer's Disease Pathogenesis’ provides a comprehensive summary of recent studies focusing on Tau protein and its interplay with the DNA damage response (DDR) in neurons. The authors explored this topic relatively in detail based on the ongoing research in their lab as well as several other published papers. This review is a summary of their research while showcasing the related research by other groups in the field.
Review is well written and the content is important. However, I have a few comments so that the review can be further improved.
-
In the abstract and conclusion the authors state that thor review aims to develop novel therapeutic strategies targeting DDR-related pathways in neurodegenerative disorders (lines 18-22, 278-280). However, the main text lacks further details in this direction. Without detailed overview of the potential therapeutic strategies the authors should refrain from anticipatory statements in review articles.
-
Section 4.1, DDR And Nuclear Membrane: The authors discussed an important experimental observation in comparison to previous studies. However, the data is not shown. It is important either to show the data, otherwise provide further details to support this claim. Please note that this is a review article.
-
Section 5.4: The sentence ‘Tau binds to microtubules and promotes microtubule polymerization, but when phosphorylated, it is released from microtubules and promotes their depolymerization’ is misleading. In fact phosphorylation is important for Tau function, whereas hyperphosphorylation leads to detachment of Tau from microtubules and hence microtubules depolymerize.
-
Section 5.4, lines 206-212: The sentences are written as presented in an original article. I recommend rewriting the sentences to better fit the review article.
-
Section 5.5: Several neurodegenerative diseases are mentioned. It would be great to provide these as sub-sections under section 5.5. Also, authors need to provide more details rather than limiting it to 2-3 sentences. Please expand this section.
-
Are there studies with molecular chaperones along with previous studies mentioned in this review? As chaperones are known to rescue Tau from pathological form is there any correlation with DDR. Actin and microtubules are the major Tau binding partners mentioned in this review. Providing the role of other Tau binding partners such as chaperones will give further insights.
-
In general, the review only briefly describes the previous research and there is room for improvement by providing additional details. For example section 5.5. Similarly other sections lack details and please try to improve providing further information.
-
Figure captions, especially figure 1: The figure is adapted from three papers, however please make sure of the permission to reuse the published figures and it should be written along with the citation.
-
Figure 2: The resolution is low, please provide a high resolution figure. Additionally, avoid highlighting the text with another color. I think it is unnecessary as it is difficult to read the content. For example, the upper middle part of the figure.
Author Response
First of all, we appreciate the time and effort the reviewers have spent reviewing this manuscript. The reviewers’ comments are very thoughtful and constructive, and we honestly agree with them. We hope the responses below address the reviewers’ comments satisfactorily.
Responses to Reviewer #1
- In the abstract and conclusion the authors state that thor review aims to develop novel therapeutic strategies targeting DDR-related pathways in neurodegenerative disorders (lines 18-22, 278-280). However, the main text lacks further details in this direction. Without detailed overview of the potential therapeutic strategies the authors should refrain from anticipatory statements in review articles.
>> Thanks for your comments. We have considered them and made significant changes. Your feedback is very important to improve our manuscript. We have rewritten the title and abstract as our data and other studies still need to allow us to present specific therapeutic strategies. The original and revised text is as follows.
P1, L2〜4
Original title: Tau Beyond Tangles: Unveiling Its Role in Neuronal DNA Damage Response and Implications for Alzheimer's Disease Pathogenesis
P1, L2
New title: Tau Beyond Tangles: DNA Damage Response and Cytoskeletal Proteins Crosstalk
- Section 4.1, DDR And Nuclear Membrane: The authors discussed an important experimental observation in comparison to previous studies. However, the data is not shown. It is important either to show the data, otherwise provide further details to support this claim. Please note that this is a review article.
>> We agree that you have not presented data on your experimental speculation about DDR and nuclear membranes. We have removed “data not shown,” added other references, and changed to a discussion of the most recent study on nuclear membrane proteins and motor protein entry into the nucleus.
P3,L112〜123
Original section 4.1
4.1. DDR And Nuclear Membrane
Nuclear invaginations are often observed in aged neurons and neurons from patients with frontotemporal dementia. Some reports suggest that these invaginations contain microtubules, nuclear membrane complexes, and phosphorylated tau that have invaded the nuclear membrane[26]. However, when we induced DSBs by X-ray exposure in a cell-based experimental system and performed 3D image reconstruction, we confirmed that the observed phenomenon is not an invagination but a widespread depression (data not shown). The previous report may have been based on a 2D reconstruction of an image of a nucleus with a depressed nuclear membrane, which might have been misinterpreted as an invagination. The nuclear membrane depression may be due to the disruption of cytoskeletal proteins scaffolding the nuclear membrane as a result of aging or DSB repair failure.
P3, L119〜P4, L167
New section 4.1. DDR And Nuclear Membrane
Nuclear invaginations are often observed in aged neurons and neurons from patients with frontotemporal dementia. Some reports suggest that these invaginations contain microtubules, nuclear membrane complexes, and phosphorylated tau that have invaded the nuclear membrane[26]. In the most recent study, Shokrollahi et al. reported that the linker of the nuclear skeleton and cytoskeleton complex (LINC) and nuclear pore complex (NPC) proteins cooperate with the nuclear lamina, kinesins KIF5B, and KIF13B to form nuclear envelope tubules to DSB sites in human cells. When DSB is induced, the DDR kinases ATM, DNAPKcs, and ATR cross with ATAT1-dependent lysine 40 acetylation of α-tubulin. The modified microtubules cooperate with the plus-end directed motor proteins kinesin-1 KIF5B, kinesin-3 KIF13B, LINC, and NPC complexes. In neurons, the kinesin superfamily is a motor protein that transports cargo proteins over microtubules and is responsible for the transport of various proteins, organelles, and mRNAs in the cell[27, 28]. Microtubules are pressed against the cytoplasmic face of the nuclear membrane and delivered to the DSB site via the resulting nuclear invasion tubular infiltrate. This tubulin-dependent nuclear invasion tubule partially promotes DSB binding, partially stabilizes microtubules, and helps direct DSBs toward the cleavage point. Because many DSB repair factors are present around the nuclear membrane, tubulin-dependent nuclear membrane entry tubes transport the nuclear membrane to the DSB and facilitate repair. Nuclear membrane entry tubes confine DSB ends within repair centers and provide strong support that promotes DNA repair protein binding and reconnection of cleaved ends [27]. Reports of phosphorylated tau entering the nucleus along with microtubules in the nuclei of FTD patient brains[26] and our results of tau accumulation around the nuclear membrane during DSB induction[15] suggest that tau may be involved in tubulin-dependent nuclear membrane entry tract dynamics in neurons, but the details are unclear.
Moreover, they observed increased 53BP1 foci following knockdown of LINC subunits, SUN1 and SUN2 [29]. It has also been reported that phosphorylated tau accumulates in NPCs in the AD brain, inhibiting nucleocytoplasmic transport [30]. Jovasevic V. et al. also reported increased nuclear membrane disruption and extranuclear release of γH2Ax with increased DSB after Contextual fear conditioning test in wild-type mice[12]. Mutations in the LMNA gene, which encodes the nuclear membrane proteins lamins A and C, cause several human diseases (laminopathies), including dilated cardiomyopathy, Emery-Dreifuss muscular dystrophy, LMNA-related congenital muscular dystrophy, and Hutchinson-Gilford premature aging syndrome [31]. Earle AJ et al. demonstrated that mechanical forces in LMNA mutant skeletal muscle cells cause myonuclear damage, nuclear membrane rupture, and DNA damage [32]. Rupture of the nuclear membrane causes DNA damage, including mislocalization of DNA repair factors and exposure of DNA to cytoplasmic exonucleases. Differentiated skeletal muscle cells downregulate DNA repair factors and are less efficient at DNA repair than undifferentiated myoblasts.
Furthermore, recent findings suggest that DNA damage and activation of the DDR pathway may be the pathogenic mechanism of myopathy [33]. Kirby TJ et al. showed increased p53 stabilization and activity in mechanically injured LMNA mutant myocytes. Furthermore, they showed that a chronic increase in p53 signaling is required to induce dysfunction in wild-type myocytes [34].
Thus, nuclear membrane depression may be due to the disruption of cytoskeletal proteins that provide scaffolding for the nuclear membrane, resulting from aging or DSB repair failure.
- Section 5.4: The sentence ‘Tau binds to microtubules and promotes microtubule polymerization, but when phosphorylated, it is released from microtubules and promotes their depolymerization’ is misleading. In fact phosphorylation is important for Tau function, whereas hyperphosphorylation leads to detachment of Tau from microtubules and hence microtubules depolymerize.
- Section 5.4, lines 206-212: The sentences are written as presented in an original article. I recommend rewriting the sentences to better fit the review article
>> Thanks for your valid comments. I deleted “Tau binds to microtubules and promotes microtubule polymerization, but when phosphorylated, it is released from microtubules and promotes their depolymerization” and added the addition about physiological phosphorylated tau and pathological phosphorylated tau.
P5,L203〜224
Original section 5.4
5.4. Microtubule Polymerization and DDR
Tau binds to microtubules and promotes microtubule polymerization, but when phosphorylated, it is released from microtubules and promotes their depolymerization [38, 39]. To investigate the effects of tau and DSBs on tubulin polymerization, we pretreated primary mouse neurons with a microtubule polymerization inhibitor and then induced DSBs with etoposide. We then determined the localization of phosphorylated tau and quantified insoluble tau. The results showed that dual exposure to microtubule polymerization inhibitors and DSB induction led to the appearance of NFT-like neurons with accumulated phosphorylated tau, increased cleaved caspase-3-positive neuronal cell death, and elevated levels of phosphorylated tau in the insoluble fraction [15]. Collectively, these findings indicate that the combination of microtubule depolymerization and DSB accumulation promotes the accumulation of phosphorylated tau in neurons. Additionally, we quantified amyloid-β in the supernatants of primary mouse neuronal cell cultures treated with varying concentrations of etoposide and found no significant changes, suggesting that DSB accumulation does not affect amyloid precursor protein (APP) cleavage [15]. These results indicate that DSB repair impairment, microtubule depolymerization, and phosphorylated tau accumulation are heavily involved in the early stages of AD pathogenesis (Figure 2). However, the mechanism of NFT formation remains unclear, and no model has yet recapitulated NFTs from endogenous tau. Elucidating the regulation of tau and its physiological mechanisms is crucial not only for understanding AD but also for deciphering the pathophysiology of tauopathies, which are pathologically characterized by tau aggregation. The mechanism of neuronal DSB repair remains to be fully elucidated.
P6,L326〜347
New section 5.4
5.4. Microtubule Polymerization and DDR
Tau is a microtubule-binding protein that localizes mainly to the axons of neurons and regulates microtubule polymerization through phosphorylation. Tau is physiologically phosphorylated and is most highly phosphorylated during the fetal stage in the mouse brain. In pathological conditions such as AD, tau is hyperphosphorylated and forms NFTs in neurons, which aggregate abnormally and cause neuronal cell death [51-53]. A study examining the effects of tau and DSBs on tubulin polymerization in primary mouse neurons reported the appearance of NFT-like neurons with phosphorylated tau accumulation and increased cleaved caspase-3 positive neuronal cell death after DSB induction following microtubule polymerization inhibition [15]. These findings indicate that the combination of microtubule depolymerization and DSB accumulation promotes the accumulation of phosphorylated tau in neurons. Additionally, we quantified amyloid-β in the supernatants of primary mouse neuronal cell cultures treated with varying concentrations of etoposide and found no significant changes, suggesting that DSB accumulation does not affect amyloid precursor protein (APP) cleavage [15]. These results indicate that DSB repair impairment, microtubule depolymerization, and phosphorylated tau accumulation are heavily involved in the early stages of AD pathogenesis (Figure 2). However, the mechanism of NFT formation remains unclear, and no model has yet recapitulated NFTs from endogenous tau. Elucidating the regulation of tau and its physiological mechanisms is crucial for understanding AD and deciphering the pathophysiology of tauopathies, which are pathologically characterized by tau aggregation. The mechanism of neuronal DSB repair remains to be fully elucidated.
- Section 5.5: Several neurodegenerative diseases are mentioned. It would be great to provide these as sub-sections under section 5.5. Also, authors need to provide more details rather than limiting it to 2-3 sentences. Please expand this section.
>> We appreciate your comments. I added more on the relationship between polyglutamine disease, PD, SCA, and DDR.
P6,L235〜P7,L268
Original Section 5.5
5.5. DDR And Pathogenic Proteins Of Neurodegenerative Diseases
Tauopathies, such as AD, are characterized by the accumulation of phosphorylated tau in neurons and glial cells. Tau is a microtubule-binding protein that localizes mainly to axons or dendrites in neurons, where it stabilizes microtubules [39]; it is also localized in the nucleus, primarily localized in the nucleolus [43-45]. In breast cancer cells, tau has been reported to bind to chromatin and contribute to chromatin integrity [40, 42] and it also chaperones the trafficking of the tumor suppressor p53-binding protein 1 (53BP1), a DSB repair factor, to DSB sites [41]. These results suggest thattau may also have a protective role in the neuronal DDR.
Synucleinopathies such as Parkinson's disease are characterized by the accumulation of phosphorylated α-synuclein in neurons and glial cells. α-Synuclein is mainly localized to presynaptic terminals of neurons and is thought to be involved in synaptic regulation and plasticity, but synuclein is also found to localize to the nucleus and has NEHJ repair capacity [46]. Conversely, overexpression of α-synuclein has been reported to activate DDR and increase DSBs [47].
Polyglutamine diseases, such as Huntington's disease and spinocerebellar ataxia, are caused by the accumulation of gene products containing expanded repeat sequences in neuronal nuclei [48]. Huntingtin, the gene responsible for Huntington's disease, has been reported to bind the NHEJ factor Ku70 [49].
The proteins implicated in amyotrophic lateral sclerosis (ALS), including TAR DNA-binding protein 43 (TDP-43), fused in sarcoma (FUS), superoxide dismutase 1 (SOD1), and chromosome 9 open reading frame 72 (C9orf72) repeat peptides, are also related to the DDR. TDP-43 is an RNA-binding nuclear protein, which stabilizes RNA [50, 51]. Additionally, a physiological function of TDP-43 related to the DDR is the recruitment of the NHEJ factor XRCC4/DNA ligase IV complex to DSB sites [52]. Furthermore, TDP43 binds to the NHEJ factor Ku70 in HEK293 cells. Knockdown of TDP-43 in HeLa cells enhances R-loop structure (Figure 1) and DSBs [53]. FUS is also involved in the transport of mitochondrial DNA ligase IIIα and base excision repair (BER) [54]. SOD1 protects against DNA damage caused by oxidative stress [53]. In HEK293 cells, C9orf72 has been reported to interact with the NHEJ factor DNA-dependent protein kinase (DNA-PK) complex after DSB induction [55].
As such, many of the proteins implicated in neurodegenerative diseases have DDR-related functions in the nucleus (Table 1). It is suggested that DDR may be present upstream as a common neurodegenerative disease mechanism.
P7,L414〜P8,475
New Section 5.5
5.5. DDR And Pathogenic Proteins Of Neurodegenerative Diseases
Tauopathies, such as AD, are characterized by the accumulation of phosphorylated tau in neurons and glial cells. Tau is a microtubule-binding protein that localizes mainly to axons or dendrites in neurons, where it stabilizes microtubules [52]; it is also localized in the nucleus, primarily localized in the nucleolus [57-59]. In breast cancer cells, tau has been reported to bind to chromatin and contribute to chromatin integrity [54, 56] and it also chaperones the trafficking of the tumor suppressor p53-binding protein 1 (53BP1), a DSB repair factor, to DSB sites [55]. These results suggest that tau may also have a protective role in the neuronal DDR. For amyloid-β, a major component of amyloid plaques and a typical AD pathology, it has been reported that administration of amyloid-β peptide to neurons and mice increases DSBs [3, 47]. In addition, the accumulation of DSBs has been reported at an early stage in AD model mouse brains where amyloid-β accumulates [40].
Synucleinopathies such as Parkinson's disease are characterized by the accumulation of phosphorylated α-synuclein in neurons and glial cells. α-Synuclein is mainly localized to presynaptic terminals of neurons and is thought to be involved in synaptic regulation and plasticity. Synuclein is also found to localize to the nucleus and has NEHJ repair capacity [60]. Conversely, overexpression of α-synuclein causes cellular senescence and activates the p53 pathway and NHEJ. However, expression of MRE11, a critical factor in the DSB repair system, is reduced, suggesting incomplete induction of the repair pathway. Neuropathological examination of α-synuclein transgenic mice showed that phosphorylated α-synuclein and DNA damage accumulate early in the presymptomatic phase, and DSB levels and cellular senescence markers are elevated [61].
Polyglutamine diseases, such as Huntington's disease and spinocerebellar ataxia, are caused by the accumulation of gene products containing expanded repeat sequences in neuronal nuclei [62]. Huntingtin, the gene responsible for Huntington's disease, has been reported to bind the NHEJ factor Ku70 [63, 64]. SCA3 is the most common form of spinocerebellar ataxia worldwide, caused by a polyglutamine (polyQ) repeat at the Ataxin-3 (ATX3) C-terminus. ATX3 has been shown to interact with polynucleotide kinase 3'-phosphatase (PNKP), mediator of DNA damage checkpoint protein 1 (MDC1), checkpoint kinase 1 (Chk1), huntingtin (HTT), Ku70, DNA-PKcs, 53BP1 and p97 [65]. Gall-Duncan T. et al. reported increased expression of the ssDNA-binding complexes canonical replication protein A (RPA1, RPA2, RPA3) and Alternative-RPA in the brains of patients with Huntington's disease and spinocerebellar ataxia type 1 (SCA1). Furthermore, overexpression of RPA in SCA1 model mouse brains reduced ATXN1 aggregation and DNA damage, restored neuronal morphology and motor phenotype, and suppressed CAG repeat expansion. [66].
The proteins implicated in amyotrophic lateral sclerosis (ALS), including TAR DNA-binding protein 43 (TDP-43), fused in sarcoma (FUS), superoxide dismutase 1 (SOD1), and chromosome 9 open reading frame 72 (C9orf72) repeat peptides, are also related to the DDR. TDP-43 is an RNA-binding nuclear protein, which stabilizes RNA [67, 68]. Additionally, a physiological function of TDP-43 related to the DDR is the recruitment of the NHEJ factor XRCC4/DNA ligase IV complex to DSB sites [69]. Furthermore, TDP43 binds to the NHEJ factor Ku70 in HEK293 cells. Knockdown of TDP-43 in HeLa cells enhances R-loop structure (Figure 1) and DSBs [70]. FUS is also involved in the transport of mitochondrial DNA ligase IIIα and base excision repair (BER) [71]. SOD1 protects against DNA damage caused by oxidative stress [70]. In HEK293 cells, C9orf72 has been reported to interact with the NHEJ factor DNA-dependent protein kinase (DNA-PK) complex after DSB induction [72].
As such, many of the proteins implicated in neurodegenerative diseases have DDR-related functions in the nucleus (Table 1). It is suggested that DDR may be present upstream as a common neurodegenerative disease mechanism.
- Are there studies with molecular chaperones along with previous studies mentioned in this review? As chaperones are known to rescue Tau from pathological form is there any correlation with DDR. Actin and microtubules are the major Tau binding partners mentioned in this review. Providing the role of other Tau binding partners such as chaperones will give further insights.
>> Thank you for your very important comments. The latest findings on kinesin, a motor protein related to axonal transport, in DSB repair and nuclear membrane invagination were added in section 4.1, “DDR And Nuclear Membrane.”
P3,L112〜123
Original section 4.1
4.1. DDR And Nuclear Membrane
Nuclear invaginations are often observed in aged neurons and neurons from patients with frontotemporal dementia. Some reports suggest that these invaginations contain microtubules, nuclear membrane complexes, and phosphorylated tau that have invaded the nuclear membrane[26]. However, when we induced DSBs by X-ray exposure in a cell-based experimental system and performed 3D image reconstruction, we confirmed that the observed phenomenon is not an invagination but a widespread depression (data not shown). The previous report may have been based on a 2D reconstruction of an image of a nucleus with a depressed nuclear membrane, which might have been misinterpreted as an invagination. The nuclear membrane depression may be due to the disruption of cytoskeletal proteins scaffolding the nuclear membrane as a result of aging or DSB repair failure.
P3, L119〜P4, L167
New section 4.1. DDR And Nuclear Membrane
Nuclear invaginations are often observed in aged neurons and neurons from patients with frontotemporal dementia. Some reports suggest that these invaginations contain microtubules, nuclear membrane complexes, and phosphorylated tau that have invaded the nuclear membrane[26]. In the most recent study, Shokrollahi et al. reported that the linker of the nuclear skeleton and cytoskeleton complex (LINC) and nuclear pore complex (NPC) proteins cooperate with the nuclear lamina, kinesins KIF5B, and KIF13B to form nuclear envelope tubules to DSB sites in human cells. When DSB is induced, the DDR kinases ATM, DNAPKcs, and ATR cross with ATAT1-dependent lysine 40 acetylation of α-tubulin. The modified microtubules cooperate with the plus-end directed motor proteins kinesin-1 KIF5B, kinesin-3 KIF13B, LINC, and NPC complexes. In neurons, the kinesin superfamily is a motor protein that transports cargo proteins over microtubules and is responsible for the transport of various proteins, organelles, and mRNAs in the cell[27, 28]. Microtubules are pressed against the cytoplasmic face of the nuclear membrane and delivered to the DSB site via the resulting nuclear invasion tubular infiltrate. This tubulin-dependent nuclear invasion tubule partially promotes DSB binding, partially stabilizes microtubules, and helps direct DSBs toward the cleavage point. Because many DSB repair factors are present around the nuclear membrane, tubulin-dependent nuclear membrane entry tubes transport the nuclear membrane to the DSB and facilitate repair. Nuclear membrane entry tubes confine DSB ends within repair centers and provide strong support that promotes DNA repair protein binding and reconnection of cleaved ends [27]. Reports of phosphorylated tau entering the nucleus along with microtubules in the nuclei of FTD patient brains[26] and our results of tau accumulation around the nuclear membrane during DSB induction[15] suggest that tau may be involved in tubulin-dependent nuclear membrane entry tract dynamics in neurons, but the details are unclear.
Moreover, they observed increased 53BP1 foci following knockdown of LINC subunits, SUN1 and SUN2 [29]. It has also been reported that phosphorylated tau accumulates in NPCs in the AD brain, inhibiting nucleocytoplasmic transport [30]. Jovasevic V. et al. also reported increased nuclear membrane disruption and extranuclear release of γH2Ax with increased DSB after Contextual fear conditioning test in wild-type mice[12]. Mutations in the LMNA gene, which encodes the nuclear membrane proteins lamins A and C, cause several human diseases (laminopathies), including dilated cardiomyopathy, Emery-Dreifuss muscular dystrophy, LMNA-related congenital muscular dystrophy, and Hutchinson-Gilford premature aging syndrome [31]. Earle AJ et al. demonstrated that mechanical forces in LMNA mutant skeletal muscle cells cause myonuclear damage, nuclear membrane rupture, and DNA damage [32]. Rupture of the nuclear membrane causes DNA damage, including mislocalization of DNA repair factors and exposure of DNA to cytoplasmic exonucleases. Differentiated skeletal muscle cells downregulate DNA repair factors and are less efficient at DNA repair than undifferentiated myoblasts.
Furthermore, recent findings suggest that DNA damage and activation of the DDR pathway may be the pathogenic mechanism of myopathy [33]. Kirby TJ et al. showed increased p53 stabilization and activity in mechanically injured LMNA mutant myocytes. Furthermore, they showed that a chronic increase in p53 signaling is required to induce dysfunction in wild-type myocytes [34].
Thus, nuclear membrane depression may be due to the disruption of cytoskeletal proteins that provide scaffolding for the nuclear membrane, resulting from aging or DSB repair failure.
- In general, the review only briefly describes the previous research and there is room for improvement by providing additional details. For example section 5.5. Similarly other sections lack details and please try to improve providing further information.
>> We agree with your comments. In particular, major revisions were made to Sections 4.1(DDR And Nuclear Membrane), 5.3 (Presence of DSBs in the AD brain), 5.4(Microtubule Polymerization and DDR),5,5(DDR And Pathogenic Proteins Of Neurodegenerative Diseases) and 6 (Conclusion).
P5,L183〜202
Original Section 5.3
5.3. Relation between DDR and Tau
When DSBs were induced in primary mouse cortical neurons using etoposide, a topoisomerase II inhibitor, non-phosphorylated tau (recognized by the Tau1 antibody, which detects tau dephosphorylated at S195, S198, S199, and S202) accumulated around the nuclear membrane and in the cytoplasm within 30 minutes of DSB induction, and its binding to microtubules also increased. Twenty-four hours after DSB induction, phosphorylated tau accumulated at the perinuclear membrane and in the cytoplasm, enhancing neuronal cell death [15]. Previous studies have shown that neuronal activity increases the localization of Top2B and calcineurin to the nuclear periphery, and induced DSB sites also localize to the nuclear membrane[20], suggesting a link to our observed DSB-induced increase in tau localization to the perinuclear membrane. To investigate the role of tau accumulation in the DDR, we performed a knockdown of endogenous mouse tau in primary mouse cortical neurons. The reduction of endogenous mouse tau increased γH2AX levels following short-term DSB induction by etoposide. However, under conditions of prolonged DSB accumulation in neurons, the knockdown of endogenous tau reduced γH2AX levels. These results suggest that tau may have a protective role in early DSB repair, but under conditions of persistent DSB accumulation, it may promote tau phosphorylation and neuronal cell death. Since neurons are thought to be unable to use homologous recombination, it is suggested that they may use NHEJ repair, which is a speedy repair. These results suggest that tau is involved in NHEJ repair.
P5,L294〜P6,L325
New Section 5.3
5.3. Relation between DDR and Tau
Tau may be associated with DDR because of its DNA-binding capacity and reports on chromatin remodeling[48, 49]. In our study results, when DSBs were induced in primary mouse cortical neurons using etoposide, a topoisomerase II inhibitor, non-phosphorylated tau (recognized by the Tau1 antibody, which detects tau dephosphorylated at S195, S198, S199, and S202) accumulated around the nuclear membrane and in the cytoplasm within 30 minutes of DSB induction, and its binding to microtubules also increased. Twenty-four hours after DSB induction, phosphorylated tau accumulated at the perinuclear membrane and in the cytoplasm, enhancing neuronal cell death [15]. The cytoplasmic enzyme GSK3β, a tau kinase that is increased in AD, is reported to translocate to the nucleus upon DSB induction, suggesting that DSBs activate a group of tau phosphorylation regulators [50]. Previous studies have shown that neuronal activity increases the localization of Top2B and calcineurin to the nuclear periphery, and induced DSB sites also localize to the nuclear membrane[20], suggesting a link to our observed DSB-induced increase in tau localization to the perinuclear membrane. To investigate the role of tau accumulation in the DDR, we performed a knockdown of endogenous mouse tau in primary mouse cortical neurons. Reducing endogenous mouse tau increased γH2AX levels following short-term DSB induction by etoposide. It has been reported that hippocampal γH2Ax is also increased in tau knockout mice [49]. However, under prolonged DSB accumulation in neurons, the knockdown of endogenous tau reduced γH2AX levels. These results suggest that tau may have a protective role in early DSB repair, but under conditions of persistent DSB accumulation, it may promote tau phosphorylation and neuronal cell death. Since neurons are thought to be unable to use homologous recombination, it is suggested that they may use NHEJ repair, which is a speedy repair. These results suggest that tau is involved in NHEJ repair.
P7,L271〜P8,L293
Original Section 5. Conclusion (Section 6 is correct).
- Conclusion
In recent years, several groundbreaking studies have been reported in the field of neurodegenerative diseases and DDR, shedding light on previously unrecognized connections. We have proposed a novel idea of DDR in NFT-mediated neuronal cell death in AD. Our findings also suggest that DDR plays a role in early DSB repair through microtubule polymerization, representing a novel physiological function of DDR in neurons. These discoveries may lead to the development of new therapeutic strategies for AD and other tauopathies, which are pathologically characterized by tau aggregation and could offer a different approach from conventional AD treatments.
Several key questions remain to be answered: "What is the mechanism that converts tau from its rapid DSB repair function to toxic phosphorylated aggregates via microtubule dynamics?" Does disruption of DSB repair lead to NFT formation?" What is the relationship between the formation of phosphorylated tau aggregates in glial cells and DSBs in other tauopathies? By elucidating physiological functions of tau that have yet to be fully characterized, we hope to explain the pathophysiology of tauopathies from the perspective of DDR.
In conclusion, this review has highlighted the intricate interplay between DDR and cytoskeletal proteins in neurons and their potential role in the pathogenesis of neurodegenerative diseases, particularly AD. The accumulation of DNA damage, especially DSBs, and the impairment of DNA repair mechanisms appear to be critical factors in the development of these disorders. The involvement of pathogenic proteins such as tau, α-synuclein, huntingtin, TDP-43, FUS, SOD1, and C9orf72 repeat peptides in the DDR suggests a common upstream mechanism in neurodegenerative diseases.
P9,L526〜540
New Section 6. Conclusion
- Conclusion
Recent groundbreaking studies on neurodegenerative diseases and DDR have been reported, shedding light on a previously unrecognized link. We proposed the novel idea of DDR in NFT-mediated neuronal death in AD. We also suggested that DDR is involved in early DSB repair via microtubule polymerization, indicating a novel physiological function of DDR in neurons. These findings have the potential to revolutionize the treatment of AD and other tau diseases that pathologize tau aggregation, leading to the development of new therapeutic strategies. These findings may provide a different approach to conventional AD treatment. Several important questions remain: "Is tau involved in tubulin-dependent nuclear membrane invasion tract dynamics in neurons?" "By what mechanism is tau converted from DSB repair function to toxic phosphorylated aggregates?" Does inhibition of DSB repair lead to NFT formation?" What is the relationship between the formation of phosphorylated tau aggregates in glial cells and DSBs in other tauopathies? We hope to explain the pathophysiology of tauopathy from a DDR perspective by elucidating the physiological functions of tau, which are not yet fully understood.
- Figure captions, especially figure 1: The figure is adapted from three papers, however please make sure of the permission to reuse the published figures and it should be written along with the citation.
>>They are also current collaborators and have agreed to reuse the NHEJ repair figures. Permission to reuse the figure was added to the Figure 1 legend.
P3,L97〜104
Original Figure 1. Legend.
Figure 1. Various NHEJ pathways in DNA double-strand break repair (DSB). Canonical NHEJ (cNHEJ) is the primary NHEJ pathway in which phosphorylated ATM phosphorylates histone H2AX at serine 139 (γH2AX) in the vicinity of DSBs, and downstream repair enzymes bind to the break ends. Alternative NHEJ (alt-NHEJ) is an alternative pathway to cNHEJ that utilizes different repair enzymes. Transcription-associated end joining (TA-EJ) is a transcription-coupled end joining pathway in which R-loop structures (hybrid DNA/RNA structures) near DSBs that arise in gene regions are protected by RAP80. Adapted from Shibata, 2017; Shibata & Jeggo, 2019; Yasuhara et al., 2022[22-24]. For detailed molecular mechanisms, see the cited references.
P3,L103〜111
New Figure 1. Legend.
Adapted from Shibata et al. with the permission of authors and publishers.
- Figure 2: The resolution is low, please provide a high resolution figure. Additionally, avoid highlighting the text with another color. I think it is unnecessary as it is difficult to read the content. For example, the upper middle part of the figure.
>>Thank you for pointing this out. Figure 2's resolution was changed to 300 dpi, and the text's highlighting was also changed, as it was difficult to see.
Please refer to P7, New Figure 2. in the body manuscript.

Reviewer 2 Report
Comments and Suggestions for Authors
The review efficiently organized an overview of (i) DDR in neurons and (i) DSBs in neurodegenerative diseases. Sections on how DSBs are associated with neuronal activity (sections 2, 4.2) and DSBs in neurodegeneration (section 5) are discussed at a good depth.
The angle of the Tau-DDR axis is not very robust. The facts and concepts localized in sections 5.3 and 5.4 heavily rely on the single article ref-15, which is a previous publication by the authors. The title of the review is strongly centered around this topic. Hence, further direct literature support is needed to claim the title topic (or the focus of the title should be changed). Points referring to the association of cytoskeletal proteins or other neurodegeneration-related proteins are good parallel facts but insufficient to support the tau-DDR connection. Similarly, the connection with Alzheimer's disease is also not clear given (and noted in Table 1) that tau is associated with multiple neurodegenerative diseases.
Some sections/statements delve into topics briefly by citing references but do not go into sufficient details. Instead, the idea finishes with a hypothesis or speculation. Such statements should be bolstered with more concrete references and logically connected statements. One such example is section 4.1, where the relation between DSB and cytoskeletal proteins is speculative and lacks clarification/reference. The same is true for lines 192-93.
References should be added to Table 1.
Overall, this is a good review of multiple good angles of discussion. However, (i) it does not focus on the title topic (ii) the supporting statements are not concrete and (iii) lacks direct references relevant to the review on this topic (which might be due to the early nature of the field).
Comments on the Quality of English Languagen/a. some spelling errors.
Author Response
Responses to Reviewer #2
The angle of the Tau-DDR axis is not very robust. The facts and concepts localized in sections 5.3 and 5.4 heavily rely on the single article ref-15, which is a previous publication by the authors. The title of the review is strongly centered around this topic. Hence, further direct literature support is needed to claim the title topic (or the focus of the title should be changed). Points referring to the association of cytoskeletal proteins or other neurodegeneration-related proteins are good parallel facts but insufficient to support the tau-DDR connection. Similarly, the connection with Alzheimer's disease is also not clear given (and noted in Table 1) that tau is associated with multiple neurodegenerative diseases. >> We agree with your precise opinion. As you pointed out, we have changed the title to one about cytoskeletal proteins and DDR because it is insufficient compared to the title focused on Alzheimer's disease, Tau, and the angle of the DDR axis.
P1, L2〜4
Original title: Tau Beyond Tangles: Unveiling Its Role in Neuronal DNA Damage Response and Implications for Alzheimer's Disease Pathogenesis
P1, L2
New title: Tau Beyond Tangles: DNA Damage Response and Cytoskeletal Proteins Crosstalk
Some sections/statements delve into topics briefly by citing references but do not go into sufficient details. Instead, the idea finishes with a hypothesis or speculation. Such statements should be bolstered with more concrete references and logically connected statements. One such example is section 4.1, where the relation between DSB and cytoskeletal proteins is speculative and lacks clarification/reference. The same is true for lines 192-93. >>We appreciate your accurate point of view. Section 4.1 “DDR And Nuclear Membrane” was mentioned with the addition of the latest paper on nuclear membrane proteins and DDR. Section 5.3 “Relation between DDR and Tau”, we added a paper on the DNA binding capacity of tau and the DSB-induced nuclear translocation of GSK3β, the phosphotransferase of tau, and discussed the DSB-induced increase in phosphorylated tau.
P3,L112〜123
Original section 4.1
4.1. DDR And Nuclear Membrane
Nuclear invaginations are often observed in aged neurons and neurons from patients with frontotemporal dementia. Some reports suggest that these invaginations contain microtubules, nuclear membrane complexes, and phosphorylated tau that have invaded the nuclear membrane[26]. However, when we induced DSBs by X-ray exposure in a cell-based experimental system and performed 3D image reconstruction, we confirmed that the observed phenomenon is not an invagination but a widespread depression (data not shown). The previous report may have been based on a 2D reconstruction of an image of a nucleus with a depressed nuclear membrane, which might have been misinterpreted as an invagination. The nuclear membrane depression may be due to the disruption of cytoskeletal proteins scaffolding the nuclear membrane as a result of aging or DSB repair failure.
P3, L119〜P4, L167
New section 4.1. DDR And Nuclear Membrane
Nuclear invaginations are often observed in aged neurons and neurons from patients with frontotemporal dementia. Some reports suggest that these invaginations contain microtubules, nuclear membrane complexes, and phosphorylated tau that have invaded the nuclear membrane[26]. In the most recent study, Shokrollahi et al. reported that the linker of the nuclear skeleton and cytoskeleton complex (LINC) and nuclear pore complex (NPC) proteins cooperate with the nuclear lamina, kinesins KIF5B, and KIF13B to form nuclear envelope tubules to DSB sites in human cells. When DSB is induced, the DDR kinases ATM, DNAPKcs, and ATR cross with ATAT1-dependent lysine 40 acetylation of α-tubulin. The modified microtubules cooperate with the plus-end directed motor proteins kinesin-1 KIF5B, kinesin-3 KIF13B, LINC, and NPC complexes. In neurons, the kinesin superfamily is a motor protein that transports cargo proteins over microtubules and is responsible for the transport of various proteins, organelles, and mRNAs in the cell[27, 28]. Microtubules are pressed against the cytoplasmic face of the nuclear membrane and delivered to the DSB site via the resulting nuclear invasion tubular infiltrate. This tubulin-dependent nuclear invasion tubule partially promotes DSB binding, partially stabilizes microtubules, and helps direct DSBs toward the cleavage point. Because many DSB repair factors are present around the nuclear membrane, tubulin-dependent nuclear membrane entry tubes transport the nuclear membrane to the DSB and facilitate repair. Nuclear membrane entry tubes confine DSB ends within repair centers and provide strong support that promotes DNA repair protein binding and reconnection of cleaved ends [27]. Reports of phosphorylated tau entering the nucleus along with microtubules in the nuclei of FTD patient brains[26] and our results of tau accumulation around the nuclear membrane during DSB induction[15] suggest that tau may be involved in tubulin-dependent nuclear membrane entry tract dynamics in neurons, but the details are unclear.
Moreover, they observed increased 53BP1 foci following knockdown of LINC subunits, SUN1 and SUN2 [29]. It has also been reported that phosphorylated tau accumulates in NPCs in the AD brain, inhibiting nucleocytoplasmic transport [30]. Jovasevic V. et al. also reported increased nuclear membrane disruption and extranuclear release of γH2Ax with increased DSB after Contextual fear conditioning test in wild-type mice[12]. Mutations in the LMNA gene, which encodes the nuclear membrane proteins lamins A and C, cause several human diseases (laminopathies), including dilated cardiomyopathy, Emery-Dreifuss muscular dystrophy, LMNA-related congenital muscular dystrophy, and Hutchinson-Gilford premature aging syndrome [31]. Earle AJ et al. demonstrated that mechanical forces in LMNA mutant skeletal muscle cells cause myonuclear damage, nuclear membrane rupture, and DNA damage [32]. Rupture of the nuclear membrane causes DNA damage, including mislocalization of DNA repair factors and exposure of DNA to cytoplasmic exonucleases. Differentiated skeletal muscle cells downregulate DNA repair factors and are less efficient at DNA repair than undifferentiated myoblasts.
Furthermore, recent findings suggest that DNA damage and activation of the DDR pathway may be the pathogenic mechanism of myopathy [33]. Kirby TJ et al. showed increased p53 stabilization and activity in mechanically injured LMNA mutant myocytes. Furthermore, they showed that a chronic increase in p53 signaling is required to induce dysfunction in wild-type myocytes [34].
Thus, nuclear membrane depression may be due to the disruption of cytoskeletal proteins that provide scaffolding for the nuclear membrane, resulting from aging or DSB repair failure.
P5,L183〜202
Original Section 5.3
5.3. Relation between DDR and Tau
When DSBs were induced in primary mouse cortical neurons using etoposide, a topoisomerase II inhibitor, non-phosphorylated tau (recognized by the Tau1 antibody, which detects tau dephosphorylated at S195, S198, S199, and S202) accumulated around the nuclear membrane and in the cytoplasm within 30 minutes of DSB induction, and its binding to microtubules also increased. Twenty-four hours after DSB induction, phosphorylated tau accumulated at the perinuclear membrane and in the cytoplasm, enhancing neuronal cell death [15]. Previous studies have shown that neuronal activity increases the localization of Top2B and calcineurin to the nuclear periphery, and induced DSB sites also localize to the nuclear membrane[20], suggesting a link to our observed DSB-induced increase in tau localization to the perinuclear membrane. To investigate the role of tau accumulation in the DDR, we performed a knockdown of endogenous mouse tau in primary mouse cortical neurons. The reduction of endogenous mouse tau increased γH2AX levels following short-term DSB induction by etoposide. However, under conditions of prolonged DSB accumulation in neurons, the knockdown of endogenous tau reduced γH2AX levels. These results suggest that tau may have a protective role in early DSB repair, but under conditions of persistent DSB accumulation, it may promote tau phosphorylation and neuronal cell death. Since neurons are thought to be unable to use homologous recombination, it is suggested that they may use NHEJ repair, which is a speedy repair. These results suggest that tau is involved in NHEJ repair.
P5,L294〜P6,L325
New Section 5.3
5.3. Relation between DDR and Tau
Tau may be associated with DDR because of its DNA-binding capacity and reports on chromatin remodeling[48, 49]. In our study results, when DSBs were induced in primary mouse cortical neurons using etoposide, a topoisomerase II inhibitor, non-phosphorylated tau (recognized by the Tau1 antibody, which detects tau dephosphorylated at S195, S198, S199, and S202) accumulated around the nuclear membrane and in the cytoplasm within 30 minutes of DSB induction, and its binding to microtubules also increased. Twenty-four hours after DSB induction, phosphorylated tau accumulated at the perinuclear membrane and in the cytoplasm, enhancing neuronal cell death [15]. The cytoplasmic enzyme GSK3β, a tau kinase that is increased in AD, is reported to translocate to the nucleus upon DSB induction, suggesting that DSBs activate a group of tau phosphorylation regulators [50]. Previous studies have shown that neuronal activity increases the localization of Top2B and calcineurin to the nuclear periphery, and induced DSB sites also localize to the nuclear membrane[20], suggesting a link to our observed DSB-induced increase in tau localization to the perinuclear membrane. To investigate the role of tau accumulation in the DDR, we performed a knockdown of endogenous mouse tau in primary mouse cortical neurons. Reducing endogenous mouse tau increased γH2AX levels following short-term DSB induction by etoposide. It has been reported that hippocampal γH2Ax is also increased in tau knockout mice [49]. However, under prolonged DSB accumulation in neurons, the knockdown of endogenous tau reduced γH2AX levels. These results suggest that tau may have a protective role in early DSB repair, but under conditions of persistent DSB accumulation, it may promote tau phosphorylation and neuronal cell death. Since neurons are thought to be unable to use homologous recombination, it is suggested that they may use NHEJ repair, which is a speedy repair. These results suggest that tau is involved in NHEJ repair.
References should be added to Table 1.
>>We agree with your comment. Reference numbers were added to “DDR-related functions” in Table 1, and ATX3 was added to “Pathogenic proteins”.
Please refer to P9,L525〜526 New Table 1 in the body mansucript.

Round 2
Reviewer 1 Report
Comments and Suggestions for Authors
I appreciate the effort of the authors to incorporate my suggestions. By rewriting the title and by including the suggested modifications, the authors have addressed my major concerns and suggestions. This is now a reasonably good review on Tau protein and its interplay with neurons' DNA damage response. I support the publication of this review in IJMS.
Author Response
Response to reviewer #1.
Comment; I appreciate the effort of the authors to incorporate my suggestions. By rewriting the title and by including the suggested modifications, the authors have addressed my major concerns and suggestions. This is now a reasonably good review on Tau protein and its interplay with neurons' DNA damage response. I support the publication of this review in IJMS.
Response: We sincerely appreciate reviewer #1's positive comment and his/her acceptance of our revision.
We hope that the latest version also meets the acceptance criteria of Reviewer #1.

Reviewer 2 Report
Comments and Suggestions for Authors
The authors have put together a good revised version of the manuscript. This is a significant modification from the previous draft. The review efficiently establishes the relationship between DNA damage repair and cytoskeletal proteins. However, the efforts to connect with tau protein remain speculative and dependent on 1-2 recent works, which does not justify the review title. Instead, the review strongly aligns with a topic such as DNA damage and cytoskeletal proteins: implication towards neurodegenerative diseases. However, that would also require incorporating more facts and elaborating DDR in neurodegeneration.
Comments on the Quality of English LanguageNA
Author Response
Responses to Reviewer #2
We appreciate the positive comments on our revision from reviewer #2. We have made some additional revisions. We hope that this revision adequately addresses the reviewer's comments.
The authors have put together a good revised version of the manuscript. This is a significant modification from the previous draft. The review efficiently establishes the relationship between DNA damage repair and cytoskeletal proteins. However, the efforts to connect with tau protein remain speculative and dependent on 1-2 recent works, which does not justify the review title. Instead, the review strongly aligns with a topic such as DNA damage and cytoskeletal proteins: implication towards neurodegenerative diseases. However, that would also require incorporating more facts and elaborating DDR in neurodegeneration.
>> We agree with your comments, so We have removed the word “Tau" from the title, abstract, and keywords and focused on cytoskeletal proteins and neurodegenerative diseases.”
P1, L2,
Original Title
Tau Beyond Tangles: DNA Damage Response and Cytoskeletal Proteins Crosstalk
P1, L2
New Title
DNA Damage Response and Cytoskeletal Proteins Crosstalk on Neurodegeneration
P1, L8〜19
Original Abstract
Neurons in the brain are continuously exposed to various sources of DNA damage. Although the mechanisms of DNA damage repair in mitotic cells have been extensively characterized, the repair pathways in post-mitotic neurons are still largely elusive. Moreover, inaccurate repair can result in deleterious mutations, including deletions, insertions, and chromosomal translocations, ultimately compromising genomic stability. Since neurons are terminally differentiated cells, they cannot employ homologous recombination (HR) for double-strand break (DSB) repair, suggesting the existence of neuron-specific repair mechanisms. Our research has centered on the microtubule-associated protein tau (MAPT), a crucial pathological protein implicated in neurodegenerative diseases, and its interplay with neurons' DNA damage response (DDR). This review aims to provide an updated synthesis of the current understanding of the complex interplay between DDR and cytoskeletal proteins in neurons, with a particular focus on the role of tau in neurodegenerative disorders.
P1, L8〜L17
New Abstract
Neurons in the brain are continuously exposed to various sources of DNA damage. Although the mechanisms of DNA damage repair in mitotic cells have been extensively characterized, the repair pathways in post-mitotic neurons are still largely elusive. Moreover, inaccurate repair can result in deleterious mutations, including deletions, insertions, and chromosomal translocations, ultimately compromising genomic stability. Since neurons are terminally differentiated cells, they cannot employ homologous recombination (HR) for double-strand break (DSB) repair, suggesting the existence of neuron-specific repair mechanisms. A central focus of this review is a potential interactions between key pathological proteins involved in neurodegenerative diseases and the neuronal DNA damage response (DDR). We aim to provide an update on the complex interplay between DDR and cytoskeletal proteins in neurons.
P1, L20
Original Keywords:
DDR; Tau; DSB; cytoskeletal protein; neurodegenerative diseases;
We appreciate the positive comments on our revision by the reviewer #2. We have made a few additional revisions. We hope this revision appropriately responded to the comments of the reviewer.
P1, L18
New Keywords
DDR; DSB; cytoskeletal protein; neurodegenerative diseases;
P2, L49〜53
Original Introduction
In this review, we present the relationship between DDR in neurons and neurodegenerative diseases, based on our findings and recent advances in the field. By exploring the intricate connections between these processes, we aim to shed light on potential therapeutic targets and strategies for neurodegenerative disorders.
P2, L47〜51
New Introduction
This review summarizes the potential role of tau protein in DNA repair and its rela-tionship to neurodegenerative diseases, presenting our published work and introduc-ing several key studies in the field. By synthesizing these findings, we aim to elucidate the complex interplay between neuronal DNA damage response (DDR) and neuro-degenerative disorders, potentially opening avenues for new therapeutic strategies.
P9, L335〜349
Original Conclusion
Recent groundbreaking studies on neurodegenerative diseases and DDR have been reported, shedding light on a previously unrecognized link. We proposed the novel idea of DDR in NFT-mediated neuronal death in AD. We also suggested that DDR is involved in early DSB repair via microtubule polymerization, indicating a novel physiological function of DDR in neurons. These findings have the potential to revolutionize the treatment of AD and other tau diseases that pathologize tau aggregation, leading to the development of new therapeutic strategies. These findings may provide a different approach to conventional AD treatment. Several important questions remain: "Is tau involved in tubulin-dependent nuclear membrane invasion tract dynamics in neurons?" "By what mechanism is tau converted from DSB repair function to toxic phosphorylated aggregates?" Does inhibition of DSB repair lead to NFT formation?" What is the relationship between the formation of phosphorylated tau aggregates in glial cells and DSBs in other tauopathies? We hope to explain the pathophysiology of tauopathy from a DDR perspective by elucidating the physiological functions of tau, which are not yet fully understood.
P9, L334〜341
New Conclusion
Recent years have seen groundbreaking studies on neurodegenerative diseases and DNA damage response (DDR), revealing previously unrecognized connections. Our work has proposed a novel concept of DDR involvement in neurofibrillary tangle (NFT)-mediated neuronal death in Alzheimer's disease (AD). Additionally, we've suggested that DDR plays a role in early double-strand break (DSB) repair via microtubule polymerization, indicating a previously unknown physiological function of DDR in neurons. These insights may offer an unconventional approach to understanding and potentially treating neurodegenerative diseases characterized by abnormal protein aggregation.
